# Decryption of Deterministic Phase-Encoded Digital Holography Using Convolutional Neural Networks

**Huang-Tian Chan and Chi-Ching Chang ***

Department of Intelligent Energy Engineering, MingDao University, Changhua 52345, Taiwan;
g5eek79@gmail.com
* Correspondence: chichang@mdu.edu.tw

**Abstract:** Digital holographic encryption is an important information security technology. Traditional encryption techniques require the use of keys to encrypt information. If the key is lost, it is difficult to recover information, so new technologies that allow legitimate authorized users to access information are necessary. This study encrypts fingerprints and other data using a deterministic phase-encoded encryption system that uses digital holography (DPDH) and determines whether decryption is possible using a convolutional neural network (CNN) using the U-net model. The U-net is trained using a series of ciphertext-plaintext pairs. The results show that the U-net model decrypts and reconstructs images and that the proposed CNN defeats the encryption system. The corresponding plaintext (fingerprint) is retrieved from the ciphertext without using the key so that the proposed method performs well in terms of decryption. The proposed scheme simplifies the decryption process and can be used for information security risk assessment.

**Keywords:** convolutional neural networks; digital holography; phase-encoded encryption





## 1. Introduction

Digital holography is used to capture and reconstruct three-dimensional (3D) images using digital sensors and algorithms [1–3]. It has applications in microscopy, biomedicine, and industrial inspection [4–6]. However, digital holography data can be sensitive and confidential so they must be secured against unauthorized access. Optical encryption is used to protect data from interception and decoding by unauthorized parties. Information security is important for finance, technology and defense. In order to ensure the security of information during transmission or storage, encryption technology must be used to process information [7–10]. Optical encryption is convenient and highly secure. It allows high-speed parallel processing of data and provides many degrees of freedom to encode the light beam, such as the phase [7,10,11], the wavelength [12] and the polarization [13]. Optical encryption began with the Double-Random-Phase Encoding (DRPE) method that was proposed by Refregier, P. and Javidi, B. in 1995 [14]. This method uses a 4f optical system and two random phase masks: one in the input plane and the other in the Fourier plane. These are used to encrypt the input image as noise. Other studies use the Fresnel Transform or Gyrator Transform for different transform domains to improve security [15,16].

There are practical risks associated with optical encryption using random phase encoding. If a decryption key is damaged or lost, the unique and irreplaceable nature of the random phase key renders the encrypted information inaccessible. Some studies use an innovative deterministic phase encoding-based holographic encryption method. G.-L. Chen et al. have conducted research on deterministic phase encoding encryption systems using digital holography [17,18]. A Lenticular Lens Array (LLA) is used as a deterministic phase mask, which has periodic phase variations, to construct an optical encryption system with established phase encoding. Using the LLA as a phase key with periodic phase encoding, the modulation of the reference light phase is controlled by the

encryptor, so there is no risk for encryption or decryption in application and the security of information is ensured.

Many common convolutional neural network (CNN) architectures are applied to deep learning for digital holography and for encryption technologies [19–21], such as U-Net, AlexNet, and ResNet. Using CNN technology for encryption allows images to be predicted without the need for an encryption key [22–24]. The use of a CNN for digital holography removes noise during image reconstruction (such as zero-order and conjugate terms) so the amplitude and phase information for digital holograms can be reconstructed directly [25–27]. If deep learning is used to reconstruct digital holograms, parameters such as object distance and light wavelength are not required to reconstruct object information without noise interference [25]. Studies show that deep learning can be used to model the relationship between ciphertext and plaintext using training samples and a trained neural network can directly predict plaintext from ciphertext. In 2020, Zhou et al. used a computer-generated hologram encryption system and a learning network to detect vulnerabilities in an optical encryption system [24]. In the same year, Zhou et al. [23] used machine learning for decryption in a study concerning optical encryption and decryption. The results show that a trained network model can extract unknown plaintext from specific ciphertext in real time, without the need to directly retrieve or estimate various optical encryption keys.

Optical encryption encodes a light beam to secure data in practical applications. However, if the encryption parameters are lost, it is difficult to reconstruct the original information. For random-phase encoding or deterministic phase encoding encryption, if a key is lost, it is not possible to decrypt the information. Currently, there is little research on machine learning that is specific to deterministic phase-encoded encryption systems, so this study proposes the use of CNN to decrypt a deterministic phase-encoded digital holographic system. The study uses a deterministic phase-encoded encryption system and computer-generated holograms (CGHs) to produce encrypted holograms, which are also known as ciphertext. The decryption process uses a U-net CNN to output plaintext.

## 2. Backgrounds

The theories and technologies for this study are described. The concept and reconstruction method for digital holography are described and the deterministic phase-encoded encryption system for this study that uses digital holography (DPDH) is presented.

### 2.1. Digital Holography

Digital holography (DH) uses the principle of interference between light waves to record the interference pattern between a known reference wave and a sample wave carrying object information. An in-line DH system is shown in Figure 1a. This uses a Mach-Zehnder interferometer and uses a laser as the light source. The light beam is guided by a mirror and is then split into object and reference waves using a beam splitter (BS) after modulation by a half-wave plate (HWP) to adjust the polarization. A polarizer (PL) is used to ensure that the object and reference waves have the same horizontal polarization. The waves are collimated plane waves after the system is set up and are captured using a CCD to produce an interference pattern. The first hologram that is recorded by the CCD is represented by Equation (1). $\psi_O$ and $\psi_R$ denote the object wave and reference wave. The intensity of the sum of the two waves is given by Equation (1), which describes the hologram $I_H$. The first two terms on the right-hand side of Equation (1) form a zero-order term, and the third term corresponds to the twin-image term, which is a virtual image and is considered as noise in the real image (fourth term).

$$I_H = |\psi_O|^2 + |\psi_R|^2 + \psi_O \psi_R^* + \psi_O^* \psi_R \tag{1}$$

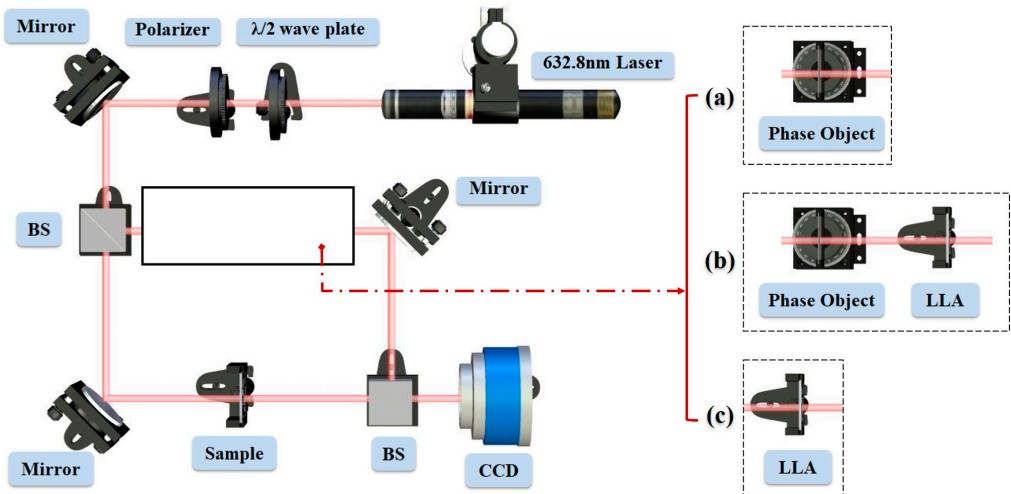

**Figure 1.** The schematic diagram illustrates an in-line digital holography (DH) system, where manipulated optical elements are represented by solid boxes. The diagram includes the following configurations: (**a**) a common in-line DH system, (**b**) a deterministic phase-encoded encryption system utilizing digital holography, and (**c**) the optical setup of the digital holographic system employed for encryption in this study.

For an in-line DH system, it is necessary to eliminate noise, such as the zero-order term and the twin-image term. Phase-shifting is commonly used to eliminate noise. Multiple holograms are produced by modulating the phase of the reference wave and the twin-image term is eliminated by computational processing. This study uses the arbitrary micro phase-step method that was proposed by Chen et al. [28]. If a phase object is placed in the path of the reference light, the optical path of the reference light is changed by rotating the phase object, which modulates the reference light phase during the recording process. If the reference wave is phase-modulated $\Delta\varphi$ by phase-shifting and the holograms that are produces before and after modulation are respectively represented by $I_{H1}$ and $I_{H2}$, they can be expressed as Equations (2) and (3):

$$I_{H1} = |\psi_O + \psi_R|^2 \tag{2}$$

$$I_{H2} = |\psi_O + \psi_R \exp(i\Delta\varphi)|^2 \tag{3}$$

If the zero-order suppression technique [29] is then used, the interference due to the zero-order term is eliminated by substituting the optical intensities of the object wave and the reference wave into the hologram. Finally, the object wave are produced by computing $I_{H1}'$ and $I_{H2}'$, which are expressed as Equations (4) and (5). The optical field distribution for the object wave is then calculated using Equation (6):

$$I_{H1}' = I_H - |\psi_O|^2 - |\psi_R|^2 = \psi_O\psi_R^* + \psi_O^*\psi_R \tag{4}$$

$$I_{H2}' = \psi_O\psi_R^* \exp(-i\Delta\varphi) + \psi_O^*\psi_R \exp(i\Delta\varphi) \tag{5}$$

$$I'_{H1} - \exp(-i\Delta\varphi)I'_{H2} = [1 - \exp(-i2\Delta\varphi)]\psi_O\psi_R^* \tag{6}$$

Digital holography involves using a computer to perform numerical computations to reconstruct a diffracted light field. In terms of the numerical reconstruction of the diffracted light field, the Huygens-Fresnel principle is used to calculate the diffracted light field $\Gamma(\xi, \eta)$ at the object plane using the conjugate light of the reference beam that passes through the hologram [30], as shown in Equation (7). The mathematical representation of the convolution calculation is denoted as $\otimes$, the function $h(\xi, \eta, x, y)$ is the impulse response of free space, d is the reconstruction distance, $(\xi, \eta)$ is the position coordinate at the object plane, and $(x, y)$ is the position coordinate at the recording plane. $\Im$ and $\Im^{-1}$

respectively denote the Fourier transform and the inverse Fourier transform, and z denotes the propagation direction of the object.

$$\Gamma(\xi, \eta) = (I_H \psi_R^*) \otimes h = \Im^{-1}\{\Im\{(I_H \psi_R^*)\}\Im\{h(z = d)\}\}. \tag{7}$$

*2.2. Deterministic Phase-Encoded Encryption Technology*

This section describes the deterministic phase-encoded encryption system that uses digital holography (DPDH) [17,18], as shown in the system architecture diagram in Figure 1b. The architecture in Figure 1b uses a lenticular lens array (LLA) as a phase modulator for the digital holography system that is shown in Figure 1a. The periodic phase variation characteristics of the LLA establish an optical encryption system that uses predetermined phase encoding. For the optical path in Figure 1b, if the reference wave passes through the LLA and propagates a distance in free space after Fresnel diffraction, the reference wave becomes the encryption key information $\psi_{key}$. For this architecture, the encryption key information and the object wave $\psi_O$ interfere, and the interference result is shown in Equation (8), which completes the encryption step:

$$I_{Encryption1} = \left|\psi_{key} + \psi_O\right|^2 = |\psi_O|^2 + \left|\psi_{key}\right|^2 + \psi_O\psi_{key}^* + \psi_O^*\psi_{key} \tag{8}$$

To decrypt the information, the zero-order interference is eliminated and the conjugate term is suppressed. The intensity of the object wave and the encrypted key information are used to eliminate the interference of the zero-order term in the digital hologram. This encrypted key information is the first key for decryption:

$$I_{Encryption1}' = I_{Encryption1} - |\psi_O|^2 - \left|\psi_{key}\right|^2 = \psi_O\psi_{key}^* + \psi_O^*\psi_{key} \tag{9}$$

The second step for decryption involves suppressing the conjugate term during the holographic reconstruction process. At this point, another encrypted hologram ($I_{Encryption2}'$) is required. The encryption hologram undergoes a phase modulation $\Delta\varphi$ with a specific amount of phase shift. In terms of the AMPS scheme [28], the conjugate term is suppressed most if the phase modulation is less than $\pi/2$. The optical field distribution is calculated using Equation (10):

$$I_{Encryption1}' - \exp(-i\Delta\varphi)I_{Encryption2}' = [1 - \exp(-i2\Delta\varphi)]\psi_O\psi_{key}^* \tag{10}$$

In terms of Equation (10), in order to obtain complete information about the object wave, the conjugate information of the key ($\psi_{key}^*$) is eliminated. Equation (7) shows that the final decryption key is the distance between the encrypted image and the CCD. To reconstruct the image, the correct distance is determined to obtain the correctly decrypted object information. In order to decrypt the encrypted image that is produced by the encryption system, various parameters, conditions and encryption keys are necessary to obtain the original information.

## 3. Experimental Design

This study uses CNN for decryption and image reconstruction. There are two stages: encryption and decryption. A phase-encoding encryption system is established that uses digital holography. Figure 1c shows the architecture of the optical system, which is established according to a previous study [17]. A 632.8 nm laser is the light source, and a polarizer and half wave plate ($\lambda/2$) are used to adjust the polarization. The beam is split by a beam splitter (BS) into two parts: a reference wave arm and an object wave arm. The object wave arm, the sample is positioned at a reconstructed distance of 15 cm away from the hologram plane. The reference wave arm has a lenticular lens array (LLA) installed at the same distance as the object wave arm, which also serves as a key. The LLA has 62 lines per inch (LPI) and the radius of curvature is 30.48 μm. Using the interference between the

two waves, the sample information is encrypted and captured using a CCD camera. During the encryption phase, experiments used two datasets: (1) Fashion-MNIST [31] and (2) the Sokoto Coventry Fingerprint (SOCOFing) Dataset [32,33]. The Fashion-MNIST dataset consists of 70,000 grayscale images that are each 28 × 28 pixels. A set of 10,000 images from the Fashion-MNIST dataset was adjusted to a size of 128 by 128. The SOCOFing dataset consists of 6000 fingerprint images from 600 African participants. All images are grayscale and have a resolution of 96 ×103 pixels. The images are cropped to 28 × 28 pixels and resized to 128 × 128. 10,000 Fashion-MNIST images and 6000 SOCOFing images were encrypted separately using the architecture that is shown in Figure 1c to obtain encrypted holograms (as ciphertext). In a phase-encoding encryption system, the decryption process uses numerical algorithms (such as the angular spectrum method, which is abbreviated to ASM) for image reconstruction. However, important information about encryption keys, such as phase objects (LLA), wavelength and distance, must be known. This study performs decryption directly through a CNN, without the need for various encryption keys.

Subsequently, in the decryption phase, a CNN is used to decrypt the ciphertexts. The CNN is U-Net, which was proposed by O. Ronneberger et al. [34]. The U-Net architecture is a commonly used CNN network architecture for digital holography, which is a widely utilized and fundamental end-to-end convolutional neural network model. It was initially designed for semantic segmentation in medical imaging and U-Net is characterized by the replacement of fully connected layers with convolutional layers. Feature extraction is performed using the down-sampling path and image restoration uses the up-sampling path. Currently, U-Net is used for image reconstruction in digital holography. The experimental setup for a U-Net model is shown in Figure 2.

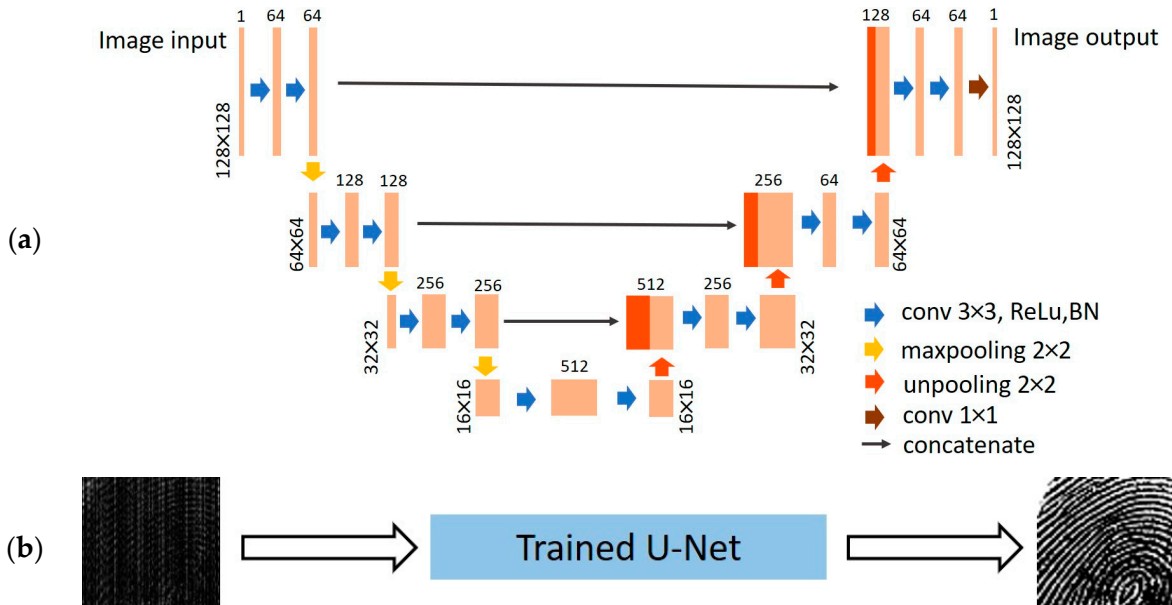

**Figure 2.** The U-Net neural network used for decryption consists of two steps: (**a**) the training step for the decryption stage, and (**b**) the testing step where the trained network is utilized to decrypt the data. The optical setup of a digital holographic system employed for encryption during the encryption stage of the encrypted data is shown in Figure 1c.

The decryption phase involves two stages: (1) training and (2) decryption. U-Net must be trained, prior to using it for decryption. A well-trained U-Net accurately predicts the output for a new input, so an encrypted hologram can be decrypted. The encrypted holograms and their corresponding original images (as plaintext) are randomly divided into three subsets, in a ratio of 70:15:15, as the training, validation, and testing sets, respectively. Using encrypted holograms (ciphertext) and their corresponding plaintext from the training and validation sets, the U-Net is trained by setting those respectively as the input and

out. After training, the U-Net is randomly assigned a ciphertext from the testing sets and outputs plaintext. The U-Net network was trained and tested using a Dell Precision 7670 workstation with an Intel Core i7-12850HX CPU (2.2 GHz) and a 16 GB RAM, using an NVIDIA RTX A1000 GPU. The U-Net network programs are written using MATLAB. The learning rate for the Adam optimizer is 0.0005 and the epoch for iterations is 10.

### 4. Experimental Results and Discussion

To verify the feasibility of the proposed method, the research methodology was tested using the Fashion-MNIST and SOCOFing databases. The first test encrypted and decrypted 10,000 images from the Fashion-MNIST database. The images were encrypted using an optical encryption system (as shown in Figure 1c). The 10,000 original images were encrypted to form holograms and were divided into training, validation, and testing sets. The U-Net network was trained using the training and validation data and the testing data was used for decryption. The testing sets were not used during the training process. The training process requires about 80 min. The RMSE and loss function plot for network training using U-Net is shown in Figure 3.

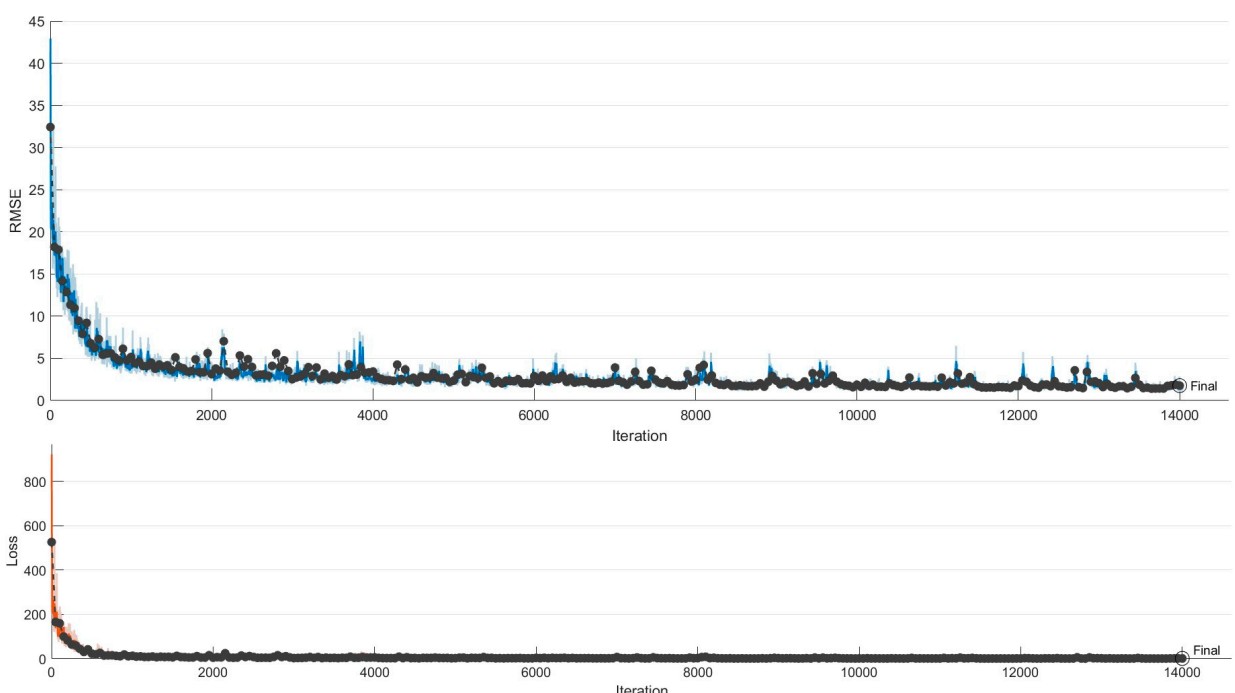

**Figure 3.** The RMSE and loss function plot for the U-Net model. The training graph displays the dataset's performance over 10 epochs.

After training, the U-Net predicts high-quality plaintext from random ciphertext inputs. The testing sets were input into the trained U-Net model. The experimental results for Fashion-MNIST are shown in Figure 4, where Figure 4a–c are the original image (as plaintext), the encrypted hologram (as ciphertext) and the decrypted image, respectively. The decrypted image (Figure 4c) can be reconstructed without knowing any information about the encryption key. U-Net is also compared with optical methods. For conventional optical setups, parameters such as wavelength, object distance, and the specifications of the encryption key (LLA) must be known. For this example, the object distance is 15 cm and encryption uses an LLA that has 62 lines per inch (LPI) as the encryption key. The reconstructed image using the correct information is shown in Figure 4d. The reconstruction results using an incorrect key and with no key are respectively shown in Figure 4e,f.

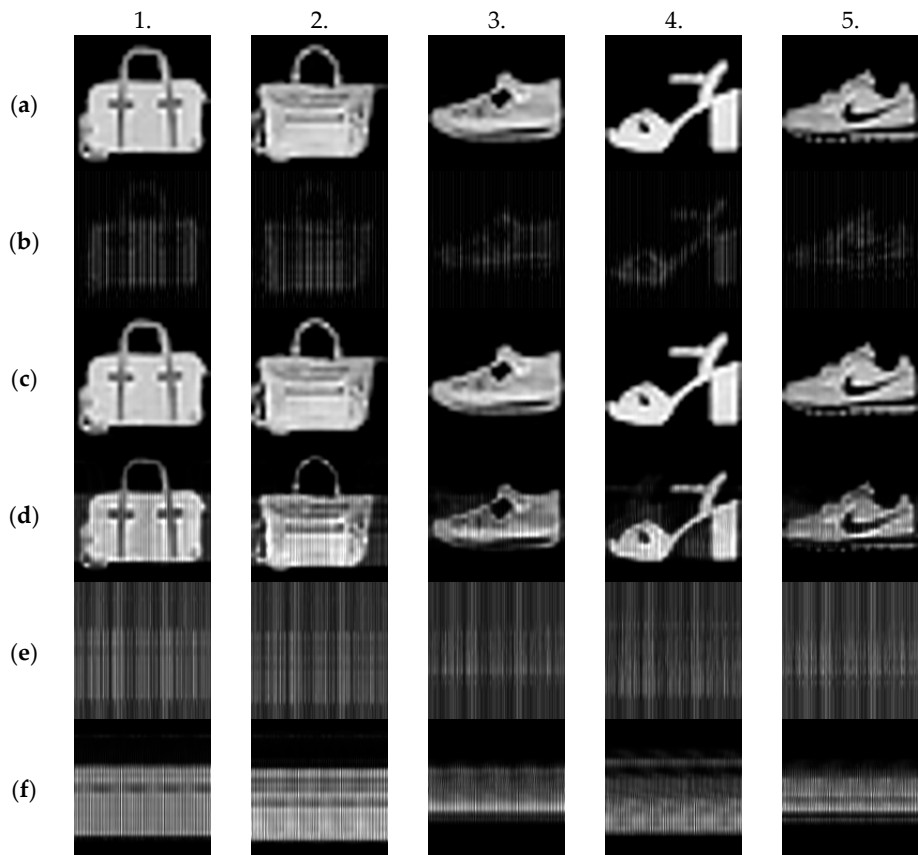

**Figure 4.** Encryption and decryption results for five samples of Fashion-MNIST data: (**a**) the original image (as plaintext), (**b**) the encrypted hologram (as ciphertext), (**c**) the decrypted image sing U-Net images deciphered using optical methods (**d**–**f**): (**d**) decrypted image using the correct key; (**e**) decrypted image using an incorrected key and (**f**) decrypted image with no key.

In order to quantitatively compare the two methods (U-Net and optical methods), the peak signal-to-noise ratio (PSNR) and the structural similarity (SSIM) between the decrypted images and the plaintext were calculated. SSIM is a metric that is used to determine the similarity between two images. The value of SSIM ranges between 0 and 1, with a higher value indicating greater similarity between two images that are compared. PSNR is a metric that is used to determine the quality of a reconstructed image. A higher PSNR value indicates that two images are very similar. The PSNR and SSIM values for the decrypted images and plaintext of U-Net and the PSNR and SSIM values for the decrypted images and plaintext using traditional techniques are shown in Table 1. The SSIM value for U-Net is 0.98–0.96, and the SSIM value for traditional decryption methods is 0.78–0.64. The results show that U-Net performs better in terms of decryption and achieves results that are closer to the original image. The image that is decrypted using U-Net also has a higher PSNR value.

**Table 1.** PSNR and SSIM values for five samples of Fashion-MNIST data.

| Method | | 1. | 2. | 3. | 4. | 5. |
|---|---|---|---|---|---|---|
| U-Net | PSNR (dB) | 36.09 | 34.26 | 38.35 | 36.80 | 36.60 |
| | SSIM | 0.96 | 0.96 | 0.97 | 0.96 | 0.98 |
| Optical methods | PSNR (dB) | 20.13 | 20.31 | 18.82 | 18.65 | 19.99 |
| | SSIM | 0.67 | 0.68 | 0.77 | 0.64 | 0.78 |

The proposed method was then verified by encrypting 6000 images from the SOCOFing database using the same approach. The U-Net network was used for training and decryption. The training process requires about 35 min. The RMSE and loss-function plot for network training using U-net are shown in Figure 5. The experimental results are shown in Figure 6. Figure 6a–c respectively show the plaintext, ciphertext and the decrypted image. The results show that the decrypted image is reconstructed without the need for information about the encryption key. Optical methods were also used to decrypt the ciphertext for comparison with the results using U-Net. The experimental results are shown in Figure 6d–f. Table 2 shows the PSNR and SSIM measurements for the decrypted images and plaintext using U-Net and using conventional decryption techniques. U-Net has a SSIM value of 0.95–0.90, so it outperforms traditional decryption methods, which typically have a SSIM value of 0.78–0.62. U-Net gives more accurate fingerprint information. The images that are produced using conventional methods have a PSNR value of approximately 15–16 dB and a SSIM value of 0.6–0.7 but using U-Net gives respective values up to 20 dB and more than 0.9.

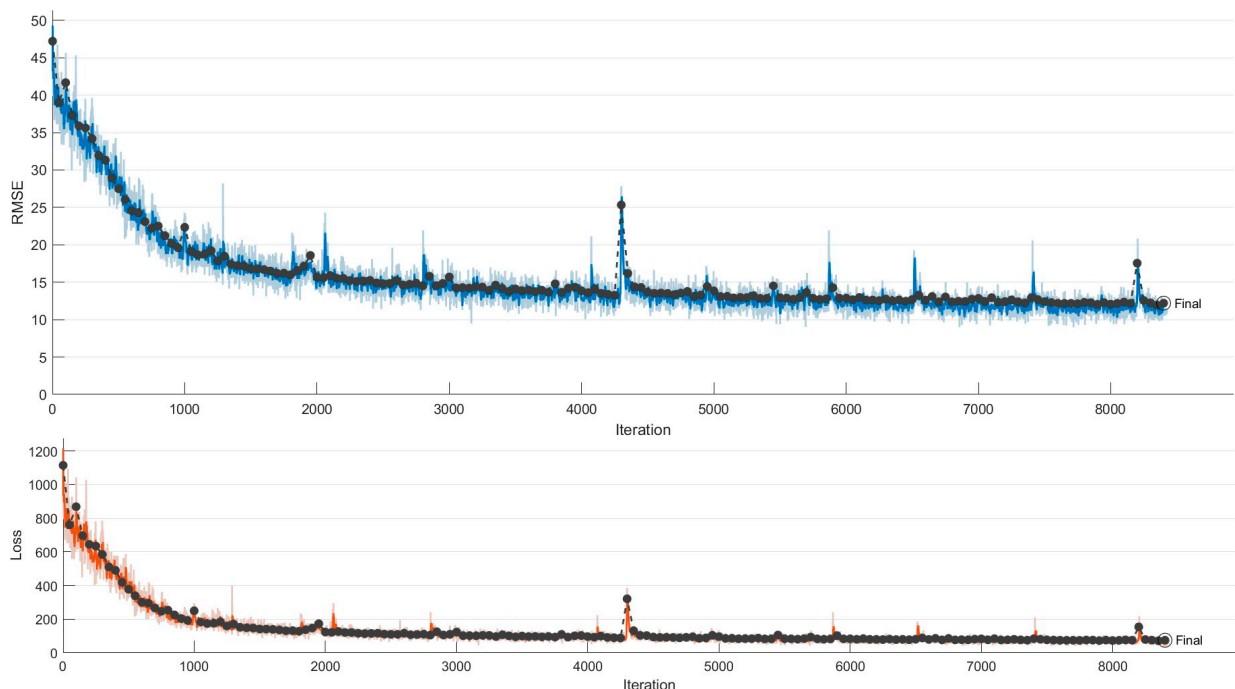

**Figure 5.** The RMSE and loss functions plot for the U-Net model. The training graph depicts the performance of the dataset over 10 epochs.

**Table 2.** PSNR and SSIM values for five samples of the SOCOFing data.

| Method | | 1. | 2. | 3. | 4. | 5. |
|---|---|---|---|---|---|---|
| U-Net | PSNR (dB) | 22.53 | 20.49 | 21.41 | 21.13 | 20.25 |
| | SSIM | 0.91 | 0.92 | 0.93 | 0.95 | 0.90 |
| Optical methods | PSNR (dB) | 16.53 | 16.44 | 15.31 | 15.36 | 16.96 |
| | SSIM | 0.62 | 0.78 | 0.67 | 0.77 | 0.78 |

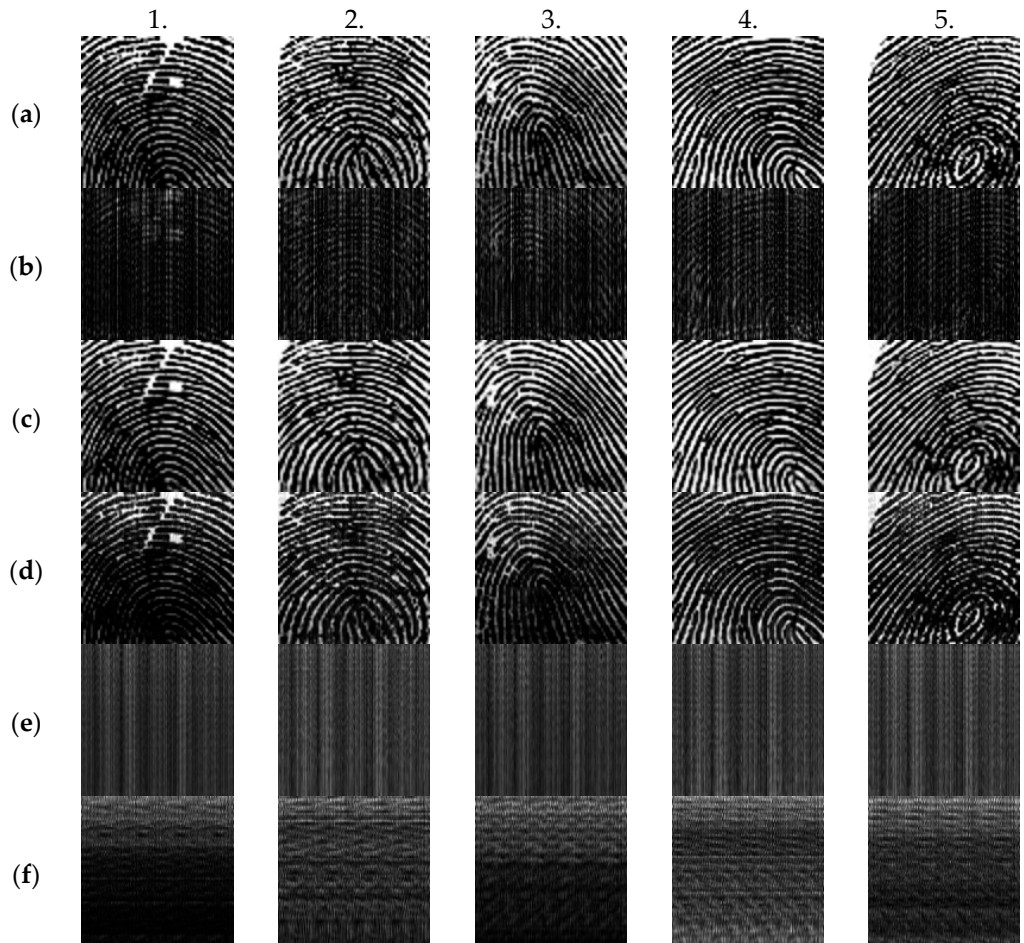

**Figure 6.** Encryption and decryption results for five samples of SOCOFing data: (**a**) the original image (as plaintext), (**b**) the encrypted hologram (as ciphertext), (**c**) the decrypted image using U-Net. Images deciphered using optical methods (**d**–**f**): (**d**) decrypted image with the correct key; (**e**) decrypted image using ab incorrect key and (**f**) decrypted image with no key.

The proposed method in this paper demonstrates the capability of a neural network to decrypt encrypted images and restore the original image without requiring parameters of the LLA, such as radius of curvature and refractive index. In real-world scenarios, it is challenging for unauthorized individuals to obtain a sufficient number of ciphertext-plaintext pairs to train a convolutional neural network for decrypting an encryption system. Moreover, these unauthorized individuals lack knowledge of the correspondence between the ciphertext and the plaintext. However, in the case of sensitive organizations such as government agencies or the financial industry, where a significant amount of important information belonging to the public is securely stored (e.g., biometric or personal data encrypted with individual consent), the authorized encryption departments naturally possess knowledge of the correspondence between the ciphertext and the plaintext. In the event of key loss due to negligence by authorized organizations, it would be possible to recover the information using CNN technology, with the permission of the individuals or consumers. In the future, two other scenarios could be explored: (1) In the Internet era, if encrypted messages lose partial information during transmission, it is worth investigating whether it is possible to restore the original image. (2) Using a structure similar to a LLA (e.g., microlens array, MA) as the encryption key.

## 5. Conclusions

This study determines the feasibility of using CNN as a decryption method. Using U-Net, the original information is directly reconstructed from a single encrypted hologram,

and the zero-order term and the twin-image term are eliminated. A trained neural network predicts plaintext directly from ciphertext. Unlike established techniques, decryption does not require parameters such as object distance, the wavelength of the light, or keys. From the perspective of consumers or manufacturers, this method is advantageous in that plaintext is obtained, even if the encryption parameters are lost. The results of this study show that U-Net performs better than traditional techniques in terms of decryption and its decrypted images are more similar to the original images. U-Net is a mature technology that can be directly applied for decryption techniques in the future.

**Author Contributions:** Conceptualization, C.-C.C. and H.-T.C.; methodology, H.-T.C.; validation, H.-T.C.; resources, C.-C.C.; data curation, C.-C.C. and H.-T.C.; writing—original draft preparation, C.-C.C. and H.-T.C.; writing—review and editing, C.-C.C.; supervision, C.-C.C. and H.-T.C. All authors have read and agreed to the published version of the manuscript.

**Funding:** The authors thank the National Science and Technology Council, Taiwan, ROC for financial support for this research under grant no. MOST 111-2221-E-451-001. The funding period for this research is from 1 August 2022, to 31 July 2023.

**Institutional Review Board Statement:** Not applicable.

**Informed Consent Statement:** Not applicable.

**Data Availability Statement:** The Fashion-MNIST and SOCOFing databases are publicly available. Related references are reported in [31–33].

**Acknowledgments:** The authors thank K. T. Li Foundation for Development of Science.

**Conflicts of Interest:** The authors declare no conflict of interest.

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
