# Peer review of "Decryption of Deterministic Phase-Encoded Digital Holography Using Convolutional Neural Networks"

_photonics, doi:10.3390/photonics10060612_

Round 1

Reviewer 1 Report

The Introduction was well structured and allowed the reader to form an understanding on the current situation in the optical encryption technology and in the deep learning. Many references were provided to articles in high-ranking journals with a publication date after 2020.

The Backgrounds section described in detail the approaches and methods that the authors proposed to use in the digital holography to the encrypt and the decrypt information.

  1. Eq.1. yo and yR  were not explicitly set in Lines 90-91 and later.
  2. Eq.7. The operators Æ‘ and Æ‘ -1 were not explicitly set, which implies the Fourier transforms. It is necessary to add a formula for the given optical operator of light propagation.
  3. Eq.7. The right side of the equation 7 “h(z=d)” contains the variable z, which is not declared on Line 114.

The Experimental Design section presented all the necessary optical system and deep learning parameters for the results repeatability.

  1. Line 159 «the of radius curvature is 30.48 m». Is this a typo or is the radius of curvature actually more than 30 meters?
  2. The authors did not sufficiently substantiate the choice of the U-Net neural network architecture in the text of the article. Comparison with other CNNs is not presented, or it is not explicitly stated that this study does not need such a comparison.

In the Experimental results and discussion section, the authors trained the U-Net convolutional neural network and compared the obtained results with conventional techniques. PSNR and SSIM are proposed as criteria for the successful implementation of the encryption and decryption algorithm.

  1. The graphs and charts of the network training process (such as: loss/epochs, accuracy/epochs) for the training and validation dataset were not presented. Adding this data would allow the readers to see in more detail the dynamics of the CNN learning.
  1. In the Conclusions or Discussions sections, it is necessary to add the information about disadvantages of the presented method, make a comparison of the presented method with the existing technologies, and describe further development prospects for this topic.

Author Response

Response to Reviewer #1

Comments and Suggestions for Authors

Summary:

This manuscript presents the use of a trained neural network to decipher phase encrypted digital holograms. The NN performs better than the optical method to restore the original image, even without knowing the parameters of the key.

We thank the reviewer for their positive assessment of our work and for acknowledging the improved performance achieved by the neural network in comparison to the optical method. We will now provide detailed responses to each of the reviewer's comments.

General comments:

Comment: The technique proposed in this manuscript is interesting and wordy of publication. It is well known that holographic phase encoding is not a 100% secure encryption technique, as it is the case for quantum cryptography. Nevertheless, research continues to be pursued into this ineffective technology. Demonstration of its vulnerability is welcome.

Response: We greatly appreciate the reviewer's interest in our proposed technique and their belief that it is worthy of publication. It is true, as mentioned by the reviewer, that holographic phase encoding is not a completely secure encryption technique when compared to quantum cryptography. We fully agree with this point. However, despite the vulnerabilities of holographic phase encoding, researchers continue to explore and study its potential. Our study primarily focuses on investigating the feasibility and limitations of holographic phase encoding. By demonstrating its vulnerabilities, we aim to enhance people's awareness and understanding of this technique. Furthermore, we encourage further research to improve the security and efficiency of holographic phase encoding. The findings of this study hold value for future work in the field of encryption and provide useful insights for researchers. We sincerely appreciate the reviewer's attention to our research and will consider these suggestions for further investigation in the future.

Comment: However, the claim from the authors that the NN can decrypt the hologram without the knowledge of the key is overblown considering the NN has been trained with pair of encrypted/decrypted images using a single key. This means the key is hard wired into the network, even though the information is not accessible. I would suggest the authors to stick to their claim that the NN was not provided with the physical parameters of the key: wavelength, focal distance and density of the lens array.

Response: We appreciate the valuable input provided by the reviewer. Based on their suggestion, we agree that it is appropriate to revise our discussion. Here is a revised version taking into consideration the reviewer's feedback (See line 278-280 in the revised MS)

We believe this revised discussion adequately addresses the reviewer's suggestion while accurately representing our findings. We appreciate the reviewer's valuable feedback in enhancing the clarity of our research.

Comment: In addition, in a real-world scenario, the eavesdropper will not have access to a significant number of encrypted/decrypted images pairs to train the network. The key can also be altered in between pairs of images. I would like the authors to add a discussion about these potential scenarios. Can a NN be trained with a significant number of key variations to decrypt an image encoded with an arbitrary key. Now, that would be impressive.

Response: Thank you for the reviewer's comment. We understand that, in a real-world scenario, the eavesdropper would not have access to a significant number of encrypted/decrypted image pairs to train the neural network (NN). The encryption key can also be altered between image pairs. While our current research focuses on the specific topic mentioned in the manuscript, we will take the reviewer's suggestion into consideration for future work. We will explore the possibility of training a NN with a significant number of key variations to decrypt an image encoded with an arbitrary key, which would indeed be an impressive achievement. Thank you for the valuable feedback, and we have incorporated it into our future research endeavors. (See line 280-294 in the revised MS)

Comment: The term “conventional technique” is not particularly explicit, and I would recommend the authors to stick to “optical method” in the document and tables.

Response: Thank you for your suggestion. We agree that using the term "optical method" instead of "conventional technique" provides better clarity and avoids any ambiguity. We will update the manuscript and tables accordingly to consistently refer to the "optical method" throughout the document. Your input is greatly appreciated, and this change does improve the overall readability and understanding of our work.

Comment: The manuscript uses a large amount of space to introduce digital holography which should already be known by the interested readers. I would encourage (although not mandatory) the authors to compress or even delete that section. I also strongly suggest to combine Figure 1, 2, and 3a in a single figure, considering the differences are minutes.

Response: We appreciate the reviewer's perspective on the introduction section of the manuscript. While it is true that digital holography may be known to some interested readers, we believe it is important to provide a comprehensive overview of the topic for readers, who may be new to the field, particularly those in the neural network (NN) domain. Understanding the fundamentals of digital holography can help readers grasp the context and significance of our proposed technique. Thank you for your suggestion to combine Figure 1, 2, and 3a into a single figure. We agree that since the differences among these figures are minimal, consolidating them will enhance the clarity and save space. We have merged the relevant content into a single figure as you recommended. (See Figure 1 in the revised MS)

Comment: Line 155-156: “Th the” please correct.

Response: We apologize for the typographical error on line 155-156. We will correct the sentence and ensure that it reads properly. (See line 159-160 in the revised MS)

Comment: Line 159: “30.48 m” seems to be missing a prefix symbol in the unit.

Response: Thank you for pointing out the missing prefix symbol in the unit "30.48 m." We will add the appropriate prefix symbol to indicate the unit correctly. (See line 163 in the revised MS)

Finally, we appreciate your valuable suggestions, and these revisions will improve the manuscript's readability and accuracy.

Reviewer 2 Report

Summary:

This manuscript presents the use of a trained neural network to decipher phase encrypted digital holograms. The NN performs better than the optical method to restore the original image, even without knowing the parameters of the key.

General comments:

The technique proposed in this manuscript is interesting and wordy of publication. It is well known that holographic phase encoding is not a 100% secure encryption technique, as it is the case for quantum cryptography. Nevertheless, research continue to be pursued into this ineffective technology. Demonstration of its vulnerability is welcome.

However, the claim from the authors that the NN can decrypt the hologram without the knowledge of the key is overblown considering the NN has been trained with pair of encrypted/decrypted images using a single key. This means the key is hard wired into the network, even though the information is not accessible. I would suggest the authors to stick to their claim that the NN was not provided with the physical parameters of the key: wavelength, focal distance and density of the lens array.

In addition, in a real-world scenario, the eavesdropper will not have access to a significant number of encrypted/decrypted images pairs to train the network. The key can also be altered in between pairs of images. I would like the authors to add a discussion about these potential scenarios. Can a NN be trained with a significant number of key variations to decrypt an image encoded with an arbitrary key. Now, that would be impressive.

The term “conventional technique” is not particularly explicit, and I would recommend the authors to stick to “optical method” in the document and tables.

The manuscript uses a large amount of space to introduce digital holography which should already be known by the interested readers. I would encourage (although not mandatory) the authors to compress or even delete that section. I also strongly suggest to combine figure 1, 2, and 3a in a single figure, considering the differences are minutes.

Line 155-156: “Th the” please correct.

Line 159: “30.48 m” seems to be missing a prefix symbol in the unit.

Author Response

Response to Reviewer #2

Comments and Suggestions for Authors

Comment: The Introduction was well structured and allowed the reader to form an understanding on the current situation in the optical encryption technology and in the deep learning. Many references were provided to articles in high-ranking journals with a publication date after 2020.

The Backgrounds section described in detail the approaches and methods that the authors proposed to use in the digital holography to the encrypt and the decrypt information.

  1. Eq.1.  and were not explicitly set in Lines 90-91 and later.
  2. Eq.7. The operators  and were not explicitly set, which implies the Fourier transforms. It is necessary to add a formula for the given optical operator of light propagation.
  3. Eq.7. The right side of the equation 7 “h(z=d)” contains the variable z, which  is not declared on Line 114.

Response: We appreciate the reviewer's positive feedback on the structure of the Introduction and the inclusion of recent references. We have carefully considered the reviewer's specific points and have made the following revisions to address them:

  1. Regarding Eq.1, and denotes the object wave and reference wave. The relevant information can be found in the range of lines 89-91 in the revised manuscript.
  2. and  denotes the Fourier transform and the inverse Fourier transform, respectively. For references on the optical operator of light propagation, please refer to the citation [30].
  3. It is true that the variable z, which represents the propagation direction of the object, was not declared in line 114 of the original MS. We make sure to explicitly declare the use of this variable in the revision (see lines 117-118 in the revised manuscript).

Comment: The Experimental Design section presented all the necessary optical system and deep learning parameters for the results repeatability.

  1. Line 159 the of radius curvature is 30.48 m. Is this a typo or is the radius of curvature actually more than 30 meters?
  2. The authors did not sufficiently substantiate the choice of the U-Net neural network architecture in the text of the article. Comparison with other CNNs is not presented, or it is not explicitly stated that this study does not need such a comparison.

Response:

We appreciate the reviewer's acknowledgment of the Experimental Design section and ensuring the inclusion of necessary parameters for result repeatability. We have carefully considered the reviewer's specific points and have made the following revisions to address them:

  1. We apologize for the confusion caused by the statement "the radius of curvature is 30.48 m" on Line 159 of the original manuscript. It was indeed a typo, and we appreciate the reviewer for catching that error. “30.48 m” should be corrected to “30.48 µm”. Amended accordingly. Please do see line 163 of the revised manuscript.
  2. The selection of U-net is due to its well-established and mature nature in the machine learning field. However, the focus of this study is currently on demonstrating that the existing U-net architecture can be used for image decryption and reconstruction. In the future, we plan to explore different CNN architectures for further research.

Comment: In the Experimental results and discussion section, the authors trained the U-Net convolutional neural network and compared the obtained results with conventional techniques. PSNR and SSIM are proposed as criteria for the successful implementation of the encryption and decryption algorithm.

  1. The graphs and charts of the network training process (such as: loss/epochs, accuracy/epochs) for the training and validation dataset were not presented. Adding this data would allow the readers to see in more detail the dynamics of the CNN learning.
  1. In the Conclusions or Discussions sections, it is necessary to add the information about disadvantages of the presented method, make a comparison of the presented method with the existing technologies, and describe further development prospects for this topic.

Response:

  1. We appreciate the reviewer's feedback on the Experimental results and discussion section. We have made the following revisions in response to the specific points raised:
  2. We have added Figures 3 and 5, which show the graphs of the network training process for the Fashion-MNIST and SOCOFing datasets, respectively. These additions provide more detailed insights into the dynamics of the CNN learning. Please refer to lines 228-231 and 267-270 in the revised manuscript.
  3. In the revised manuscript, we have included information about the disadvantages of the presented method, a comparison with existing technologies, and discussed further development prospects for this topic. Specifically, we address the challenges of obtaining sufficient ciphertext-plaintext pairs for training a CNN in real-world scenarios. Additionally, we highlight the potential for information recovery in the event of key loss with proper authorization. We also mention two future scenarios involving the restoration of partially lost encrypted messages during transmission and the use of a structure similar to a LLA (e.g., microlens array, MA) as an encryption key. Please see lines 280-294 in the revised manuscript for the relevant information.

We appreciate the insightful suggestions provided by the reviewer. By incorporating these revisions, we believe that the manuscript will provide a more comprehensive and informative discussion, improving the overall quality of the paper.
